# Biomimetic Hyaluronan Binding Biomaterials to Capture the Complex Regulation of Hyaluronan in Tissue Development and Function

**DOI:** 10.3390/biomimetics9080499

**Published:** 2024-08-17

**Authors:** Amelia Huffer, Mingyang Mao, Katherine Ballard, Tugba Ozdemir

**Affiliations:** Nanoscience and Biomedical Engineering Department, South Dakota School of Mines, Rapid City, SD 57701, USAmingyang.mao@mines.sdsmt.edu (M.M.); katherine.ballard@mines.sdsmt.edu (K.B.)

**Keywords:** hyaladherin, HA binding peptides, biomaterials, tissue regeneration, therapeutic

## Abstract

Within native ECM, Hyaluronan (HA) undergoes remarkable structural remodeling through its binding receptors and proteins called hyaladherins. Hyaladherins contain a group of tandem repeat sequences, such as LINK domains, B_x_B7 homologous sequences, or 20–50 amino acid long short peptide sequences that have high affinity towards side chains of HA. The HA binding sequences are critical players in HA distribution and regulation within tissues and potentially attractive therapeutic targets to regulate HA synthesis and organization. While HA is a versatile and successful biopolymer, most HA-based therapeutics have major differences from a native HA molecule, such as molecular weight discrepancies, crosslinking state, and remodeling with other HA binding proteins. Recent studies showed the promise of HA binding domains being used as therapeutic biomaterials for osteoarthritic, ocular, or cardiovascular therapeutic products. However, we propose that there is a significant potential for HA binding materials to reveal the physiological functions of HA in a more realistic setting. This review is focused on giving a comprehensive overview of the connections between HA’s role in the body and the potential of HA binding material applications in therapeutics and regenerative medicine. We begin with an introduction to HA then discuss HA binding molecules and the process of HA binding. Finally, we discuss HA binding materials anf the future prospects of potential HA binding biomaterials systems in the field of biomaterials and tissue engineering.

## 1. Introduction

Hyaluronic acid (HA) is a versatile extracellular matrix molecule utilized heavily as a therapeutic in the literature [1]. While the existing data are very promising and there is evidence showing the relationship between the molecular weight of HA and its pro- or anti-inflammatory properties [2], it is not fully clear how different tissues react to HA with different molecular weights and concentrations. One of the most striking differences between therapeutically used HA and native HA is its crosslinking state. The native HA is usually present in remarkably high molecular weights (2–9 MDa) that are highly crosslinked with other HA binding proteins and HA interacting cell surface receptors [3]. The molecular structure of HA is rather simpler with repeating disaccharide units; thus, the diversity in HA binding molecules points to the diverse functions possessed by HA. Translating the HA binding and crosslinking mechanisms into material design has been the focus of promising biomaterials designed for improving outcomes of diverse pathological conditions. Understanding the importance of HA in tissue organization and function, depicting the mechanisms of complex HA binding interactions, and translating them into biomaterial designs for developing future HA binding materials are the focus of this article. We first introduced the importance of HA in tissue development and structural organization in Section 2. We followed by explaining different HA binding molecules and how these molecules diverge in different tissues and tissue functions in Section 3. In Section 4, we revised the existing HA binding biomaterials applied in the regeneration of different tissues and discussed potential future HA binding biomaterial strategies. We finished the review with a commentary on our projection on how future HA binding materials can be developed for diverse tissue regenerative purposes.

## 2. Regulatory Roles of HA in Tissue Function and Development

HA is a non-sulfated glycosaminoglycan expressed by three enzymes, hyaluronan synthase 1, 2, and 3 (HAS1, HAS2, and HAS3). HA forms long linear chains of polymeric macromolecules both inside and outside of cells [4]. The chain length of HA (which is the molecular weight), is tightly regulated by hyaluronidases and environmental physical signals (ionic strength, stiffness, or tension). This change in chain length has significant biological outcomes in regeneration, healing, or disease progression. Unlike complex proteins, like collagens or elastin, HA has a relatively simple structure [2]. A single HA monomer consists of a disaccharide of D-Glucoronate-1,3—N-acetylglycosaminoglycan at each repeating unit. Cells can recognize HA through its binding receptors (Figure 1). CD44, the receptor for hyaluronic acid mediated motility (RHAMM), LYVE-1, HARE/Stabilin, and Toll-like receptors (TLRs) are transmembrane cellular receptors for HA. Despite their simple structure, HA forms complex binding interactions with a variety of hyaluronan interacting proteins, including but not limited to aggrecan, versican, neurocan, tenascin, biglycan, and inter-alpha-trypsin inhibitor (IaI). As part of the extracellular matrix, HA attracts water, which leads to increased hydration and lubrication and regulates the stiffness of the extracellular space. The molecular weight of HA has important cellular functions; high molecular weight (HMWHA > 900 kDa) is shown to have anti-inflammatory effects and low molecular weight (LMWHA < 120 kDa) is shown to have pro-inflammatory effects by regulating cell proliferation, migration, and differentiation in different tissues [5]. The distribution, crosslinking interactions, and molecular weights of HA change drastically between tissues during health, regeneration, and disease. In this section, we will discuss the roles of HA in different tissues’ development and function.

### 2.1. Role of HA in Skin Tissue Development and Function

Human fetal skin development follows three stages: organogenesis, histogenesis, and maturation. During the organogenesis (0–60 days) stage, the ectoderm that is lateral to the neural plate is specified to become an epidermis and a subset of mesenchymal and neural crest cells become the dermis [6]. At this stage, the primitive epidermis and primitive dermis initiate paracrine crosstalk for basement membrane and skin appendage (hair, nail, and sweat gland) development. During histogenesis (60 days to 5 months), primitive skin undergoes substantial morphological changes, including stratification, appendage involution and mesenchymal subdivision of the epidermis and dermis, and neovascularization [7]. During the maturation (5–9 months) stage, the functional evolutions of specialized skin components develop, such as thermoregulatory, surface tensile strength, and barrier function. In most fetal tissues, HA exhibits a gel-like, highly hydrated matrix that gives space and elasticity to the fast-proliferating and migrating embryonic cells. High concentrations of HA are present in the early stages of embryonic skin differentiation and, as the tissue morphogenesis continues, both the levels of HA and CD44 drop. With the accompanying loss of HA and CD44, epithelial cells accumulate and lead to the first morphological signs of the induction of skin appendages. It is also well documented that the wound created on the gestational fetus healed without scars [8]. Several differences exist between adult wound healing and fetal wound healing. Strikingly, the levels, order of assembly, molecular weight, and HA receptor expressions exhibit significant differences in fetal wounds. Upon excision or injury to fetal skin, the first extracellular matrix deposited in the wound bed is HA. The molecular weight of fetal wound HA is a high molecular weight and higher in concentration than adult wounds, leading to a diminished and shorter inflammatory stage. Finally, it is found that the HA receptors are also highly expressed in fetal dermal cells, implicating more binding interactions with the embryonic HA [6].

The majority of the HA present in the body is located in our skin. Both the concentration and MW of HA show significant changes with age. More specifically, human skin contains the highest HA concentration at embryonic stages and, upon birth, we lose almost 50% of the HA in the first 6 months. As we age, both concentrations and the MW are shown to decrease, losing the supple, hydrated structure of the skin. HA is present in both epidermal and dermal compartments of skin [4]. In the epidermis, HA is located at the basement membrane underneath the stratum basale, as well as the pericellular space between the adjacent keratinocytes on all other epidermal compartments (stratum corneum, stratum lucidum, and stratum granulosum and stratum spinosum). Keratinocytes express high levels of CD44 and form both reversible and irreversible bonds with HA [9]. The literature found that CD44-HMWHA interactions usually form irreversible bonding, primarily regulating HA assembly, while CD44-LMWHA interactions form reversible bonds, leading to the endocytosis of HA, and further intracellular signaling events, leading to cell proliferation [10]. In addition to keratinocyte function, resident Dendritic/Langerhans cells also express high levels of surface CD44 and, upon injury, migrate near the keratinocyte bed, bind to HA, and further mature. This event leads to decreased Toll-like receptor (TLR) TLR2/4 levels at the cell membrane, which has been shown to aid skin wound healing. In the dermis, the predominant cell types regulating dermal organization are fibroblasts, hair follicle cells, resident stem cells, and resident innate and adaptive immune cells [11]. Fibroblasts produce the majority of the skin’s HA, with amounts of 400–800 mg/g dry weight making skin the third most HA-dense tissue (after joint and eye) [12]. Upon injury, HA is the first macromolecular ECM component synthesized by fibroblasts. Swelling of HA in the ECM prevents the crowding of other ECM components, regulates collagen organization, and controls neovascularization and immune function [4]. The stiffness of the microenvironment is regulated by the HA and we have shown earlier that stiffness can impact both healthy and diseased fibroblasts [13]. HA has been shown to play roles at all stages of wound healing (hemostasis, inflammation, proliferation, and remodeling). In the hemostasis stage, HA in serum regulates platelet aggregation and fibrin mesh formation. In the inflammatory stage, HA interacts with both innate and adaptive immune cells. For instance, during skin injury, CD44-expressing mast cells elevate interleukin-10 expression (IL-10), which, in turn, activates fibroblasts to produce excessive HMWHA and diminish collagen deposition and inflammatory macrophage polarization. The molecular weight of HA also has implications for immune response in skin injury. For instance, LMWHA induces the polarization of the pro-inflammatory macrophage (M1) via increasing their TLR2/4 levels while HMWHA induces the polarization of the anti-inflammatory macrophage (M2). The HMWHA also interacts with regulatory T cells (Treg) through elevated expression levels of IL-2 and immunosuppressive IL-10 and TGF-b [10] (Figure 2).

### 2.2. Role of HA in Cartilage Tissue Development and Function

Cartilage and synovium undergo various structural and functional changes after birth. The articular cartilage is isotropic and highly cellularized at birth but as the tissue matures, unique zones develop. This distinction of unique zones allows the articular cartilage to withstand high compressive and shear forces throughout the range of motion for a joint. The compact region of mesenchymal cells is called the interzone and animal studies have found that removal of this zone from embryos results in halting the formation of limb joints. The superficial zone is found at the surface adjacent to the joint cavity and is made up of elongated, flattened cells parallel to the surface. These cells are important in creating HA and lubricin to maintain the frictionless movement of a joint. The appearance of this region is the first sign of joint development. During aging and response to injury, these vital components can be compromised [14]. HA plays a major role in the ECM and is found in articular cartilage and synovial fluid (SF). HA is produced by chondrocytes and synoviocytes. The HA produced plays a role in the biomechanics of synovial fluid, specifically in the viscoelasticity of the SF and lubrication. The concentration and MW of the HA in the cartilage decrease with age and with diseases like osteoarthritis [15]. Over the course of aging, hyaluronidases degrade the high-molecular-weight HA to proinflammatory low-molecular-weight HA [16].

Articular cartilage is a nearly frictionless, load-bearing, lubricating surface that supports and distributes forces generated during loading and movement in diarthrodial joints. The ECM of this cartilage is made up of chondrocytes. Chondrocytes are surrounded by a thin pericellular matrix (PCM). Not much is known about the functions of the PCM but it is believed to serve as a transducer of both biochemical and biomechanical signals for the chondrocytes. The matrix molecules type VI collagen, perlecan, aggrecan, HA, and type IX collagen are found either exclusively or at higher concentrations in the PCM compared to the surrounding ECM. These matrix molecules and the interactions between them contribute to the mesh-like structure of the PCM and the biomechanical properties [17]. The role of pericellular HA in articular cartilage is very similar to the role of aggrecan. HA retains the aggrecan within the cartilage, ultimately supporting aggrecans’ biomechanical function. The anchoring of aggrecan to the cell surface can only occur by binding HA to the aggrecan and the HA is tethered to either the CD44 or HA synthase (HAS) [18]. HAS2 is the predominant isoform involved in the synthesis of HA in human cartilage. HA in cartilage is responsible for aggrecan retention within the PCM and this contributes to the establishment and maintenance of cell–cell spacing distances between chondrocytes. In some studies, it was found that when catabolism was enhanced by the use of serum-free conditions, the half-life of aggrecan and that of HA were similar, which led to the conclusion that HA turnover was in some part mediated by endocytosis by chondrocytes in articular cartilage [18].

Synovium is the soft tissue lining the spaces of tendon sheaths, diarthrodial joints, and bursae. It is derived from ectoderm and does not contain basal lamina. The main functions of synovium are lubrication of cartilage, maintenance of an intact non-adherent tissue surface, control of the synovial fluid volume and composition, and nutrition of chondrocytes within joints. The intima (continuous layer of cells) and the underlying tissues (the subintima) are the two main layers within the synovium. Diffusion of HA from the surface toward clearing lymphatics may be indicated by the decrease in HA levels in the deeper subintima layer. Large amounts of HA are found in the intima and superficial subintima layers of the synovium. The HA level begins to decrease in the deeper subintima layer, which may indicate diffusion of HA from the surface toward clearing lymphatics. Type B synoviocytes have adapted to the production of HA. Specifically, CD68-intima fibroblasts demonstrate the high activity of UDP-glucose dehydrogenase, which converts UDP-glucose into UDP-glucuronate, which is one of the two substrates needed by hyaluronan synthase for the assembly of HA. The HA and lubricin help maintain the lubrication and volume of synovial fluid near the cartilage surface during exercise. It is believed that the mechanical stimulation of intimal fibroblasts and the effectiveness of synovial fluid cushioning help control the rate of synthesis of HA and its exportation into the synovial fluid compartment. For example, when the synovial fluid volume is high, the mechanical stress on the fibroblasts is reduced, lowering the rate of HA production (Figure 2). When more friction is present in exercise, there is more HA produced [19].

### 2.3. Role of HA in Musculoskeletal Tissue Development and Function

The differentiation of embryonic stem cells (ESCs) into smooth muscle cells (SMCs) involves dynamic changes in HA metabolism. During muscle differentiation, which begins as early as Day 3, there was a notable upregulation in HAS expression, leading to a substantial accumulation of extracellular HA and its organization into pericellular coats [20]. While differential regulation was observed for all HAS genes (HAS1, HAS2, and HAS3), only HAS2 consistently correlated with myogenic differentiation. Depletion of HAS2 hindered differentiation, resulting in a loss of cell-associated HA and the HA-dependent pericellular matrix. This inhibitory effect on differentiation due to HA loss was further validated using the HA synthesis inhibitor 4-methylumbelliferone [21]. Notably, HAS2 exhibits elevated expression levels in the distal posterior subridge mesoderm of the chick limb bud while being absent from the apical ectodermal ridge (AER). Consequently, an increasing proximal-to-distal gradient of HA is observed in the progress zone, which correlates with the cellular differentiation state along this axis [22]. The HA-depleted region is postulated to facilitate physical interactions between the AER and the underlying mesenchymal cells. Furthermore, the CD44 localization may be involved in sustaining a HA-free zone [23] through endocytosis and subsequent intracellular degradation of HA [24,25].

HA plays a crucial role in bone turnover regulation [26]. While links between bone resorption and the induction of HAS in osteoclasts have been established, these findings suggest that HA may contribute to the binding of osteoclasts to bone surfaces and the formation of “sealing zones”, both of which are essential processes for bone resorption [27]. Some early studies of the oral administration of HA showed a reduction in bone resorption [28]; however, the contribution of osteoclasts has not yet been determined [26,29]. In vitro studies have explored the effect of HA on osteogenesis by examining various parameters, including colony number, osteoblast proliferation, osteocalcin mRNA levels, and alkaline phosphatase activity [30,31,32]. Although preliminary, some investigations suggest that HA promotes osteogenesis depending on its molecular weight and concentration. Notably, HMWHA has been reported to exhibit osteoinductive properties in vivo [33,34]; however, this observation contrasts with another study demonstrating the failure of HA to promote bone formation in distraction osteogenesis [35]. As other glycosaminoglycans might also influence bone, the osteoinductive effects of HA require further investigation and caution should be exercised when describing their specificity [36,37].

HA is abundant in almost every tissue and constitutes 8% to 10% of the total dry mass in the human body. Muscle tissue is also very rich in HA content. HA is found to be present in different compartments of muscle tissue, such as endomysium, perimysium, and epimysium, which are the connective tissue sheaths enveloping individual muscle fibers, fascicles, and the entire muscle, respectively. During muscle contraction, HA plays a crucial role in facilitating lubrication and lateral force transmission, thereby contributing to the efficient mechanical functioning of skeletal muscles [38,39]. Thus, maintaining HA homeostasis is crucial for muscle physiological functions. Deviations from optimal levels, whether decreased or increased, can significantly impact the musculoskeletal system, leading to pathological outcomes. An overabundance of HA within the ECM of muscle can substantially augment viscosity and modify its lubricative characteristics. While ECM viscosity has been overlooked as a factor contributing to passive resistance in muscle, it is found that ECM viscosity is significantly elevated due to muscle hyperactivity [40]. Consequently, the heightened passive resistance to movement and diminished force transmission may culminate in muscle stiffness [41,42]. Moreover, HA is presented in minor quantities within the fluid of serous cavities, such as the pleura, pericardium, and peritoneum, as well as in the less-defined tissue planes facilitating movement between muscle bodies and skin. The distribution of HA is homogenously interspersed within individual fibers, spindles, and septa in cardiac and skeletal muscle while diminishing significantly in slow-moving smooth muscle fibers of the gut and vessel walls [43].

Despite the long-standing knowledge that bone organ cultures can produce HA, the function of HA in bone remains unclear. A key factor contributing to this gap in understanding is the absence of a well-defined structural role for HA in bone, in contrast to its pivotal role in cartilage. This lack of an apparent structural function in bone has hindered efforts to elucidate the functional significance of HA in this tissue [26].

HA has been proposed to play a potential role as a regulator of mineralization in bone [26]. While HA does not directly influence mineral growth, its high binding affinity for hydroxyapatite (the principal mineral component of calcified cartilage and bone) may affect a regulatory influence in mineralization [44]. Notably, HMWHA has been shown to increase osteoblast proliferation and mineralization, whereas LMWHA (60 kDa) enhances proliferation without affecting mineralization [30]. These findings suggest that the molecular weight of HA may differentially modulate the processes of osteoblast proliferation and mineralization (Figure 2).

### 2.4. Role of HA in Eye Development and Function

During embryonic development, the eyes form around the third to tenth week of gestation. The tissue of the eye has mesodermal and ectodermal origin. At the sixth week of gestation, the development of the vitreous humor begins and HA can be seen throughout the entire development. Additionally, HA is also found in the developing retina but in lower amounts [45]. The concentration in the vitreous humor decreases as we age and the mechanical properties of the vitreous humor change and the concentration of HA decreases. These changes can lead to different pathologies that are commonly seen in people as they age [46]. During aging, there is a decrease in the concentration of HA that is present in the vitreous humor and this can lead to a stiffening of the gel due to dehydration [47].

Limbal epithelial stem cells (LESCs) are found in the corneal limbus in the basal layer of the corneal epithelium. The LESCs are important in maintaining the corneal epithelium by differentiating into new corneal epithelial cells after an injury. The LESC ECM is rich in HA and when there is a disruption in this environment, there is a delay in wound healing. After injury, HAS1 is most active and responsible for producing HA [48]. The LSCs will co-localize with the HA in the matrix and this indicates that LSCs have a direct interaction with HA [49].

The eye is a sensory organ which is responsible for vision. The ECM of the eye is a gel-like substance called the vitreous humor that fills the space between the lens and the retina. The vitreous humor provides support to the eye and helps maintain the shape of the eye and it is comprised of various collagens, stabilin, fibronectin, laminin, proteoglycans, and HA (Figure 2). HA in the vitreous humor fills spaces between collagen fibrils and provides pressure that inflates the gel. Within vitreous humor, HA is not found to be distributed homogeneously. The highest concentration of HA is found in the posterior vitreous cortex [46]. HA is made by orbital fibroblasts. These orbital fibroblasts have all three different isoforms of hyaluronan synthases (HASs). HAS2 is the main contributor of HA synthesized by the fibroblasts. HA synthesis is stimulated by various signals, including inflammation and adipogenesis [50]. The tear film surrounds the cornea and separates the surface of the eye from the environment. Its functions are to maintain comfort, prevent infection, suppress inflammation, and clear debris. HA is one of the main components of the tear film and it aids in providing stability. HA has viscoelastic properties that allow there to be moisture surrounding the eye at all times. The viscosity of the tear film decreases during a blink, which allows for an even distribution of the tear film. HA also has many hydroxyl groups that will attract water molecules, aiding in the thickening and stabilization of the tear film. HA also aids in preventing the evaporation of the tear film. These effects of HA cause adequate lubrication at the ocular surface [51].

HA is also found in the cornea and has several functions. It is responsible for maintaining the epithelial stem cell niche and helps in wound healing after an injury [48,49,52]. After an injury, HA is upregulated to aid in the healing process. Similar to other tissues in the body, there are HA binding proteins, such as fibronectin and CD44, which are also found in the eye, and the presence of HA causes the expression of these proteins to increase. The interaction between HA and these proteins contributes to the promotion of corneal epithelial migration and the re-epithelization of the wound [52,53]. Although HA promotes the migration of corneal epithelial cells, it does not stimulate the proliferation of epithelial cells [52,53]. During the wound healing of the cornea, HA molecules of different molecular weights have different roles in immune cell activity. HMWHA will inhibit the activity of immune cells, such as the phagocytosis of macrophages and neutrophils. Conversely, LMWHA is an activator of macrophages and it induces several cytokines, such as TNF-alpha, as well as chemokines [54].

### 2.5. Role of HA in Neural Tissue Development and Function

We can divide the role of HA during development into three parts. First, the role of HA in early development has been detected in the very early stages of development; in particular, neural tube folding is directly associated with the involvement of HA. The chick embryos treated with hyaluronidases failed the folding of neural tubes, proposing HA’s structural support and tensile strength during neural tube folding and closure [55]. Further studies conducted on chick embryos suggest HA is also critical in both early neurulation and spinal cord formation [56]. In addition to the neural tube folding, the presence of HMWHA also contributes to the migration of neural crest cells [57].

Second, In the later stages of brain and central nervous system development, HA appears to be present in several regions of the developing mouse brain, including but not limited to the cortex, cerebellum, striatum, and subpallidian structures [57,58]. The presence of a highly hydrated HA-rich matrix in the developing brain is hypothesized to provide a cell-free matrix for neuronal precursor migration and the radial and tangential movement of cells in the rapidly developing brain [59]. In addition, the HA-CD44 or HA-RHAMM interactions appear to regulate the maturation of neural structures, such as lamination and sprouting in the hippocampus [60] or the sorting of axons, such as CD44+ axons in optical chiasm [61] and RHAMM+ axons in noradrenergic neurons in corpus coeruleus [62].

Finally, HA has important roles in neural stem cell maintenance, differentiation, and perineural network (PNN) formation [63]. In developing frog brains, HA synthases and HA receptors are found to be highly concentrated at the proliferative ventricular zones where embryonic neurogenesis takes place. In addition, the binding of HA-RHAMM and HA-CD44 is closely associated with mitotic spindle formation and orientation [64,65]. For instance, the binding of HA-CD44 at the membrane is shown to impact mitotic spindle orientation in neuronal stem cells controlling the decision to undergo asymmetric cell division for the maintenance of stemness [65]. HA interacts with versican, neurocan, brevican, and aggrecan to form tightly knit perineural networks at the interstitial space.

The roles of sulfated and non-sulfated glycosaminoglycans are comparatively important in both neural function and development. In the adult nervous system, these proteoglycans and unique protein ligands form a PNN, which constitutes about 20% of the neural tissue. The PNN consists of a loose network of primarily HA, sulfated glycosaminoglycans, and tenascin-R [66]. The interactions between Fibronectin domains on tenascin-R and lectin domains on hyalectans and, finally, hyalectan binding to HA create giant macromolecular structures that create lubricated, highly hydrated, and structurally stable ECM around neurons. Especially aggrecan, which is one of the hyalectans in the PNN and is very rich in neural ECM [67,68]. In healthy neuronal ECM, HA performs as a molecule that determines the interactions between other neural ECM molecules. HA is synthesized via HAS 1–3 in the neural ECM and released through the plasma membrane into the ECM by astrocytes and localized around both astrocytes, dendritic cells, and other neuronal cells [57]. Other hyalectans in the nervous system can be listed as brevican, neurocan, and versican [69]. In addition to tightly organized HA-Hyalectin-tenascin-R networks, there are small leucin-rich proteins present in neural ECM, such as biglycan and decorin. Research indicates the levels of these small leucine-rich proteins (SLRPs) significantly increase upon injury, as well as LMWHA production, which could last at least 6 months after injury [70]. In maintaining neuronal homeostasis, HMWHA decreases inflammation by blocking the ability of binding of ECM ligands to innate immune receptors, such as TLR4. In vitro, cultured astrocytes treated with HMWHA showed diminished proliferation and chondroitin sulfate proteoglycan (CSPG) deposition. During or after neural tissue injury, the highly organized HMWHA-rich matrix underwent significant degradation and, alongside HA degradation, synthesis of a new tenascin molecule (tenascin-C) hallmarks the changes in neural tissue. Tenascin-C acts as a damage-associated molecular pattern (DAMP), which leads to the activation of several other matrix metalloproteases, which leads to chronic inflammation after neural tissue injury (Figure 2).

### 2.6. Role of HA in Cardiovascular Tissue Development and Function

HA is crucial in orchestrating the characteristic transition of cardiac endothelial cells to mesenchymal cells, a pivotal developmental process in cardiovascular tissue formation [71]. In mice lacking HAS2, the extracellular matrices surrounding embryonic cells exhibit increased density, impeding cell migration and leading to severe cardiac and vascular deformities, ultimately culminating in midgestational demise [72]. This defect is reversible through gene rescue, HA supplementation, or expression of constitutively active Ras mutants. Conversely, deficiency in Hyal2 results in HA accumulation and interstitial cell buildup, contributing to congenital heart defects and subsequent heart failure [73].

Developing valves are rich in HA, which helps maintain a physiological environment for tissue remodeling without retraction [74]. HA exerts multiple functions during atrioventricular canal (AVC) morphogenesis [72]. HA is essential for the formation of the expanded, acellular ECM, which facilitates the migration of endocardial cells. Additionally, it provides a migratory substrate for these invasive cells and promotes Ras-dependent transformation into pre-valvular mesenchyme. An enzyme studied in zebrafish produces UDP-glucuronate, a necessary precursor for HA synthesis by HAS2. Collectively, these observations indicate that the presence of HA is indispensable for the normal morphogenetic processes of chamber septation and heart valve formation [72]. HA fragments, especially oligomers, have been observed to stimulate endothelial cell (EC) proliferation [75] and promote angiogenesis [76], which represents the new growth of blood vessels. However, these effects are often influenced by concomitant signaling from other growth factors. Notably, HA oligomers have been shown to compete with native long-chain HA in binding to the HA receptor CD44 present on ECs. In contrast to long-chain HA, the HA oligomers trigger intracellular signaling pathways that promote active cell proliferation [77].

In cardiovascular ECM, glycosaminoglycan (GAG) polymers comprise chondroitin sulfate, HA, heparan sulfate, and keratan sulfate, with sulfated variants typically binding to proteoglycan core proteins before secretion. Chondroitin sulfate proteoglycans, such as aggrecan and versican, interact with HA and cartilage link with Protein 1 to form aggregates, contributing to ECM hydration [78]. The accurate synthesis and structural arrangement of the ECM within endocardial cushions play a pivotal role in valve development and maturation. HA stands as the predominant GAG in the developing heart, playing a crucial role in promoting valvular endothelial cell proliferation and facilitating Endothelial-to-Mesenchymal Transition (EndMT) (Figure 2). Deficiencies in HA resulting from either loss-of-function mutations or excessive degradation in HAS2 and described Ugdh genes, which are crucial for HA and other GAG syntheses, are implicated in cushion and valve defects [79,80]. Conversely, an excess of HA leads to the onset of excessive EndMT. HA is known to interact with the erythroblastic oncogene B2 (ErbB2)/ErbB3, serving as the endocardial signaling receptor for neuregulin1, a key player in the regulation of endocardial EndMT and cushion ECM synthesis. Similarly, the architecture of the HA matrix network relies on the integration of HA binding proteins, or hyaladherins, such as versican. Additionally, HA crosslinking can occur via tumor-necrosis-factor-stimulated Gene 6 and Pentraxin 3 [81].

### 2.7. Role of HA in Immune System Development and Function

Cellular components of the immune system originate in the bone marrow hematopoietic stem cells and HA is the major extracellular component of bone marrow. HA in the bone marrow not only provides structural and physical support but also interacts with the receptors of HA binding proteins to affect the cellular functions of hematopoietic stem cells. HA removal by hyaluronidase in bone marrow resulted in the demolishing of both bone marrow progenitors and mature cells, indicating that HA is an essential component for both the maintenance of stemness and differentiation of HSCs [82]. In addition to cellular components, another important component of the immune system is the lymphatic vessels and lymph nodes. The majority of HA that is produced in the body is either turned over or degraded in lymph nodes and transported via lymphatic vessels. Lymphatic vessels carry an exclusive HA receptor LYVE-1 (lymphatic vessel endothelial HA receptor) [83]. During lymphatic system embryonic development and fate determination, LYVE-1 is one of the earliest surface antigens of newly developed lymphatic vessels, indicating the importance of HA in lymphatic system development. Thus as the system develops and matures LYVE-1 interacts actively with leukocyte surface antigens to transport HA to the lymph nodes [83]. HA and HA binding proteins play significant roles in immune function. We mentioned earlier in the skin section about the host defense mechanisms involving HA however the role of HA is not limited to only wound healing [84]. The majority of evidence regarding HA–immune-system interactions points to CD44-mediated cellular responses; however, cytokine release, even in the CD44^−/−^-mice, suggests other regulators that may be playing roles in the inflammatory response. HA is a simple carbohydrate molecule with repeating disaccharide residues showing structural similarity with pathogenic cell membrane disaccharides, making it an appealing ligand for Toll-like receptors (TLRs) [85]. For instance, TLR4 and TLR2 are two of the important TLRs recognizing the lipopolysaccharide repeat units and secretion of chemokine activity. For instance, HA representation is completely abolished in TLR4^−/−^ and TLR2^−/−^ macrophages [86]. HA oligosaccharides also lead to the maturation of dendritic cells via TLRs. The HA oligosaccharide representation increased TNF-α production and NF-_K_B nuclear translocation, leading to dendritic cell maturation. TLR4 and 2 also help in modulating alloimmunity through dendritic cell maturation [87]. While it is thought that all immune cells carry HA binding receptors and one way or the other interact with HA, some immune cells exhibit specific interactions in response to HA of varying molecular weight and concentration. T cells can only respond to HA when they are activated; thus, naive T cells did not show HA binding ability due to the decreased or non-existent membrane CD44 [88]. When activated, it is noted that CD44 levels significantly increase the activated circulating T cells exhibited rolling over vascular endothelial secreted HA within the glycocalyx. In addition to circulating T lymphocytes, skin resident gamma-kappa T cells show a significant role in keratinocyte growth and HA expression [89]. In addition, CD4+ CD25+ regulatory T cells’s immunosuppressive function depends heavily on the molecular weight of the HA fragments present in the extracellular milieu [90]. In addition to T cells, both neutrophils and macrophages interact with HA via their cell surface CD44 molecules [91]. In polymorphonuclear neutrophils (PMNs), the cell polarization and directed migration depend heavily on the CD44-HA interactions. In addition, the ligation of CD44 through anti-CD44 antibodies led to IL-6 secretion and, furthermore, irreversibly crosslinking CD44 on PMNs led to apoptosis [92]. The effect of HA and its fragments is heavily studied in macrophage maturation and differentiation. While high-molecular-weight HA leads to an anti-inflammatory M2 phenotype, degraded low-molecular-weight fragments of HA lead to a pro-inflammatory M1 phenotype, suggesting a clear distinction in macrophage response to HA [93]. This response is predominantly regulated by CD44-HA interactions (Figure 2). While the majority of immune cell interactions with HA led to cellular migration, proliferation or cytokine/growth factor secretion interactions of HA with mast cells led to the secretion of HA itself by the granulosomes in the mast cells and deposition in the extracellular space [94]. For instance, patients with interstitial cystitis were treated with intravesicular HA derived from activated mast cells due to its possible replenishments of bladder glycosaminoglycans [95].

### 2.8. Role of HA in Reproductive System Development and Function

Hyaluronan (HA) is important in the development of the reproductive system in both males and females. Research has found that HA also plays a pivotal role in fertility and the ability of a person to reproduce [96]. In males, HA affects the sperm structure and physiology. It has been found that modification of HA in sperm membranes can influence their motility and improve their ability to bind. In females, HA plays a role in the development of ovarian follicles and the formation of cumulus-oocyte complexes. The concentration of HA is found to be lower in ovaries compared to other urogenital organs. The cumulus-oocyte complex (COC) is made up of many layers of cumulus cells compacted together and surrounding the maturing oocyte. During the process of cumulus expansion, the synthesis of HA in the cells regulates the expression of HAS2 [96]. Ultimately a HA-rich matrix is created in the COCs. The aspect of ECM helps interconnect the oocyte and the cumulus cells, which helps protect the mechanical factors throughout ovulation. This helps lead to better chances of a successful pregnancy.

In the reproductive system, the enzymatic activity of hyaluronan synthase genes HAS1 and HAS2 are known to synthesize the HA. In fetal development, it is essential for HA to bind to CD44 during cumulus expansion, which leads to maturation-promoting factor (MPF) activity and oocyte meiosis [97]. It has also been found that HA may have a big role in enhancing the ability of an embryo to adhere to the endometrial surface of the uterus due to HA’s ability to promote adhesion between cells and the ECM. For a more in-depth understanding of the relationship between HA and reproductive biology, Ghaleno, L.R. et al. have a review article on the subject [96].

From a fertility standpoint, there has been much research into the ability of HA to improve fertility in patients. There have been links found between HA levels and the ability of sperm to survive in the vagina until reaching the egg cell to fertilize (Figure 2). As previously mentioned, there is also a belief that HA levels may also aid in the attachment ability of the oocyte to the endometrial tissues during implantation. This can also be extended to creating better outcomes for artificial insemination and other types of reproductive interventions [97].

### 2.9. Role of HA in Respiratory Tissue Development and Function

The ECM of lung tissue is made up of components, including elastin, collagen, and glycosaminoglycans. HA is a key component in the ECM of respiratory tissues and is found in high levels in the connective tissue of the lungs. Research has found that HA plays a key role in the regulation of water in the interstitium. This is due to the water-retaining property of the HA and can also play a role in regulating the water levels in the tissues [98].

In the airway, HA is primarily found in the pulmonary vasculature walls, the airway submucosa, and in lower amounts in the alveoli. There have been many studies looking into the effectiveness of HA as a therapeutic for respiratory disorders. It has been found that for respiratory disorders, like chronic sinitus, HA is more effective at preventing reoccurring infections when compared to current treatments lacking HA [99,100].

When lung tissue is damaged, it produces large quantities of short, fragmented HA, which is known to have proinflammatory and proangiogenic effects. Other studies have found a correlation between elevated levels of HA in bronchoalveolar lavage (BAL) fluid and environmental and occupational airway injuries. There were also findings that suggest the presence of elevated HA levels that may occur before the injury to the tissue may be found, such as in the case of the formation of fibrosis within the lung tissue.

## 3. HA Binding Proteins and Discovery of HA Binding Peptides

Hyaluronan (HA) is found in several different tissues and it has a variety of functions. HA’s ability to have such diverse functions is due to its ability to interact with a wide range of HA binding peptides and tandem repeat protein regions in hyaladherins. Several HA binding proteins exist both in the extracellular space and as the HA receptor at the cell membrane. In this section, we will first briefly introduce each HA binding protein with described functions in the literature. We will then talk about how those proteins interact with HA through specialized binding domains; we also provided a comprehensive table (Table 1) describing the tissue distribution, function, and binding mechanism of each HA binding protein we could find drafting this article.

As a linear chain polymer with significant water solubility, the HA seldom exists alone in the native ECM. HA interacts heavily with a multitude of cell membrane receptors (CD44, RHAMM, LYVE-1, TLRs) and ECM molecules (i.e., aggrecan, brevican, neurocan, versican, TSG-6, stabilin). HA and HA binding protein interactions often exist through either LINK modules or BX_7_B motifs where B is arginine or lysine and X is any non-acidic amino acid [122,128]. Through these interactions, HA remains as a stiffened and highly dynamic molecule that is constantly forming semi-ordered states [129]. In its simplest form, HA colonizes the extracellular space towards forming periodic filamentous complexes (owing to the repeating nature of the polysaccharide) depending on the conformation of the bound sugar and the nature of the stabilizing protein–protein associations [84]. These complex associations serve not only as protective mechanisms to initiate inflammatory responses, directional migration, or cellular proliferation but also to inhibit the loss of ECM components during those protective mechanisms mentioned above [84]. Furthermore, depending on the location and the nature of the interactions HA and its binding interactions could elicit significantly different biological responses.

### 3.1. HA Binding Proteins (Hyaladherins)

HA can interact with both cell membrane receptors and both intra and extracellular scaffolding proteins (Figure 1). Below we summarize each protein that directly interacts with HA.

#### 3.1.1. CD 44

CD44 is a common HA receptor that is found in various tissues throughout the body. It is involved in several cellular processes. CD44 expression levels are higher in cancer cells (Table 1). The interaction between CD44 and HA activates different signaling pathways, including MAPK, which can lead to an increase in tumor cell proliferation and migration [130]. During inflammation, CD44 is upregulated and is found on the surface of T cells. CD44 is involved in the migration of T cells through its interactions with vascular endothelial cells allowing T cells to enter the bloodstream [131]. 

#### 3.1.2. RHAMM

RHAMM is another common HA receptor that is found throughout the body in various tissues. RHAMM and HA interactions have been found to increase cell motility, which has implications for tissue repair and cancer cell motility (Table 1). In cancer cell migration, interactions between RHAMM and HA activate the ERK1/2 pathway. Activation of this pathway might lead to epithelial cells changing their morphology to be more mesenchymal-like [132]. RHAMM and HA interactions can also regulate the integrity of the mitotic spindle, which can also impact the motility of the cells [133]. RHAMM has also been used in phage display studies to extract HA binding peptide sequences [134].

#### 3.1.3. LYVE-1

LYVE-1 is a HA receptor that is found in lymphatic vessels. HA enters the lymphatic vessels and then goes to the lymph nodes where it is degraded before it enters circulation [135]. LYVE-1 interactions can affect the integrity of the lymphatic vessels by causing the junctions of the vessels to become relaxed (Table 1). LYVE-1 and HA interactions are also known to be involved in the process of leukocyte trafficking where the immune cells adhere to migrate across the vascular endothelial to go to a site of inflammation [136].

#### 3.1.4. Toll-like Receptors

Toll-like receptors (TLRs) are pattern recognition receptors that span through the cell membrane of a wide range of immune and non-immune cells (monocytes, macrophages, dendritic cells, neutrophils, B cells, T cells, fibroblasts, endothelial cells, and epithelial cells. In humans, 10 functional TLRs have been reported and TLR2 and TLR4 have shown interactions with HA directly and indirectly [137]. The role of HA in TLR signaling is dependent on molecular weight, primarily revolving around the LMWHA. While several studies showed HA can bind directly to TLR2 and TLR4 [99,138,139], there is a hypothesis that not all TLR2 or TLR4 cells bind to HA but instead, degradation of HMWHA around the pericellular coat makes the HA available for TLR signaling to get activated [140].

#### 3.1.5. Stabilin

Stabilin is a multifunctional transmembrane receptor and takes roles in a variety of cellular processes, such as angiogenesis, lymphocyte homing, cell adhesion, and receptor recycling (Table 1). Stabilin both binds to HA and endocytose for processing.

#### 3.1.6. TSG-6-TNFIP6

TSG-6 is found in the ECM of various tissues and is involved in the inflammatory process (Table 1). The expression of TSG-6 is influenced by several factors, including TGF-beta and TNF-alpha. TSG-6 can also enhance the binding of HA to the cell surface of CD44 on lymphoid cell lines. It was also found that TSG-6 complexes can cause CD44 to switch into an active state [141]. This can lead to enhancement in the effects that occur when CD44 and HA interact.

#### 3.1.7. Aggrecan

Aggrecan is a major component in cartilage but can also be found in both the cardiovascular and the nervous system [142]. Aggrecan interacts with growth factors and morphogens that direct tissue morphogenesis, remodeling, and metaplasia. Specifically, the human natural killer trisaccharide (HNK-1) aggrecan glycoform is neuroprotective in PNNs in the brain and direct neural crest cell migration in embryonic development (Table 1). Aggrecan is found in the central nervous system (CNS) and peripheral nervous system (PNS) in the PNN structures. The structures are aggrecan-HA-tenascin C aggregates, which localize around neurons during development. This is also considered a specialized form of neural ECM that has neuroprotective roles and controls synaptic plasticity. Aggrecan has also been shown to play a role in cartilage development and it has been found that there is an increase in sulfated aggrecan with tissue maturation. In articular cartilage specifically, aggrecan forms macro-aggregate structures through interactions of its N-terminal G1 domain with HA and LINK protein [143].

#### 3.1.8. Brevican

Brevican is the smallest proteoglycan of the lectican family and is considered the most abundant proteoglycan in the ECM of the CNS. Brevican contains a HA binding domain in its N-terminus and a lectin-like domain at the C-terminus [144]. It is secreted by astrocytes and neurons and is believed to inhibit plasticity after injury [145]. While not much is known about the mechanisms involving brevican, studies have found a connection between Brevican and glioma cell motility, adhesion, and tumor growth (Table 1). Malignant gliomas exhibit a unique brevican isoform and the brevican is critical for its proinvasive role in gliomas. Studies have shown that brevican is overexpressed in patients who have these gliomas [146].

#### 3.1.9. Neurocan

Neurocan is another member of the lectican family, which is known to form large aggregates with HA and tenascins in which many signaling molecules and enzymes, including matrix proteases, are preserved (Table 1). Secreted chondroitin sulfate proteoglycans, like neurocan, are highly expressed in the areas that are strategically important for neuronal migration in the development of the cerebrum [147].

#### 3.1.10. Versican

Versican is found everywhere in the body but also has been shown to have a reciprocal effect on aggrecan when found in the same tissues (Table 1). Versican is known to regulate various biological processes, like vasculogenesis, cell proliferation, and differentiation during development [142]. Versican is known to interact with both HA and the HA receptor CD44 (Figure 3). This plays a role in heart failure as the CD44 pathway plays a role in heart failure and it has been found that versican accumulates in patients with ischemic heart failure [101].

#### 3.1.11. HABP1/C1QBP

Hyaluronan binding protein 1 (HABP1) or complement component 1 q subcomponent binding protein (C1QBP) is a HA binding protein originally isolated from brain and liver samples [148,149]. HABP1 is an intracellular protein located in mitochondria and golgi and shown to be distributed homogenously during mitosis reflecting HA distribution inside the cell. HABP1 that is bound to HA interacts with ERK1 inside, which involves cell proliferation [150]. Overexpression of HABP1 in rat skin fibroblasts showed extensive vacuolation and reduced growth rate and eventually underwent apoptosis [151].

#### 3.1.12. HARE

Hyaluronan receptor for endocytosis (HARE) is a receptor specifically present in sinusoidal endothelial cells of the liver, lymph nodes, and spleen [152,153]. The HARE encoding gene contains several BX_7_B HA binding motifs [154]. HARE mediates systemic clearance of HA from both systemic and lymphatic vessels via coated pit-mediated uptake [155]. There are different isoforms of HARE and most of the HARE isoforms are intracellularly located within the endocytic system, indicating a transient surface residency for endocytic recycling receptors [156]. This specificity makes HARE not only a functional HA endocytic receptor but also a marker for the identification of liver sinusoidal endothelial cells. 

#### 3.1.13. SHAP

SHAP refers to serum-derived HA-associated proteins and this protein molecule is covalently bound to HA through ester linkages [157]. SHAP appears to mediate the binding of HA to cell surfaces and may represent one of the serum factors implicated in HA metabolism in cultured fibroblasts [158]. Also, it is plausible that SHAPs may constitute the two heavy chains of the inter-alpha-trypsin inhibitor (ITI), but without the light chain. These proteins could potentially participate in the formation of a HA-rich extracellular matrix through their calcium-dependent HA binding activity [159].

#### 3.1.14. SPACR

SPACR serves as a significant constituent of the retinal interphotoreceptor matrix. It is a proteoglycan believed to contribute to the preservation of photoreceptor cell viability and facilitate adhesion between the neural retina and the retinal pigment epithelium. Furthermore, SPACR enables binding with HA, heparin, and chondroitin. Moreover, SPACR is an important composition of ECM [125]. 

#### 3.1.15. HA Binding Peptides (HABPs)

There are several different types of HA binding peptides. One group of peptides all have a domain called a LINK domain and this domain is involved in ligand binding [129]. The LINK domain was first described in aggrecan, a protein found in cartilage. This protein has two LINK domains right next to each other [160]. The LINK module has a very similar structure to the C-type lectin domain. There are differences in the mechanisms involved in carbohydrate binding but the locations of ligand-binding surfaces are very similar. They also have common roles which indicates that they have a common evolutionary origin. The C-type lectin domain is present in many proteins of invertebrates whereas the LINK module has only been found in vertebrates. This indicates that the LINK module evolved from the C-type lectin domain. It is also thought possible that the LINK module and HA could have appeared at about the same time during evolution [129].

RHAMM is another common HA binding protein. This protein is unique because it contains a linear sequence of amino acids responsible for HA binding, which is not seen in other HA binding proteins. It was found that the HA binding domain of RHAMM is a region of about 35 amino acids and it is near the carboxyl terminus of the protein [122].

### 3.2. HA Binding Domains Located on HA Binding Molecules

#### 3.2.1. LINK Domains

The LINK domain is found in many hyaluronan binding proteins and is involved in the binding of HA to the protein (Figure 4). This domain has approximately one hundred amino acids. Within this domain, there are four cysteines that are bound together with a disulfide bond in the pattern of Cys1–Cys4 and Cys2–Cys3. The three-dimensional structure has been determined using nuclear magnetic resonance spectroscopy and it revealed that the three-dimensional structure is made of two α-helices and two triple-stranded anti-parallel β-sheets. This structure is known as the consensus fold. This fold is highly conserved across proteins that are a part of the LINK module, especially in regions that correspond to parts of the secondary structure. There are three distinct types of LINK domains that are based on the size of their HA binding domain, which correlates to the minimum size of HA that is recognized by the protein. Type A LINK modules have an HA binding domain of about ninety amino acids, and the binding of HA is regulated by pH. TSG-6 is an example of a Type A LINK module. The type B LINK module has a HA binding domain of about one-hundred and sixty amino acids with two extensions. CD44 and LYVE-1 are examples of proteins that have Type B LINK domains. The third type of LINK module is Type C, which consists of two LINK molecules involved in binding. This is seen in proteins, such as aggrecan and versican. The minimum HA size recognized by a Type C LINK module is larger than that of a Type A LINK module. The amino acids that make the LINK module are not found next to each other when the protein is in its primary structure. This indicates that the correct secondary structure depends on the correct folding of the protein [129].

#### 3.2.2. RHAMM-Derived HA Binding Domains and Their Roles

RHAMM is part of a larger group of proteins called hyaladherins, proteins that can bind to HA. These proteins are commonly found in the ECM, on cell surfaces, and intracellularly. While many proteins in this group have a LINK domain consisting of one-hundred amino acids, RHAMM, on the other hand, binds to HA via a linear binding motif called BX_7_B, which consists of only nine to eleven amino acids [129,134]. RHAMM also does not contain a peptide or a transmembrane domain, so it leaves the cell through chaperone proteins and combines with the cell surface. Most proteins containing a LINK domain are intracellular proteins but RHAMM can be found intracellularly and at the cell surface. RHAMM can be found in different forms, with the largest being an 85 kDa intracellular protein, but there are also smaller variants. RHAMM also consists of five functional domains and Domain 5 (D5) is where the HA binding motif can be found. HA binding to this domain is essential for the motility signal to be sent to the cells. It appears that cell surface RHAMM will have HA directly binding to D5 whereas intracellular RHAMM uses D5 to bind to extracellular-signal-regulated kinase (ERK). The largest form of RHAMM is considered inactive and the shorter variants are considered active. In HA’s presence, there is a rapid increase in the expression of shorter RHAMMs (Table 1). For example, after a skin injury to a rat, there was a large increase in the expression of RHAMM but, after 24 h, RHAMM expression had diminished. In diseased tissue, RHAMM expression is more constant while in healthy tissue, RHAMM expression comes and goes [134].

#### 3.2.3. CD44-Derived HA Binding Domains and Their Roles

CD44 is a common HA receptor (Table 1). Throughout different tissues, CD44 can be found in many different isoforms due to the splicing of 10 different exons in a variety of combinations. For instance, in the chondrocyte pericellular region, transmembrane CD44 binds to the HA linear chain while aggrecan extends HA structural modulation in the pericellular region right beneath the CD44-HA binding locus. All the CD44 variants contain a LINK domain near the N-terminus, which is involved in HA binding and is part of the LINK module superfamily. It is one of only two members of the LINK module superfamily identified as a HA receptor alongside LYVE-1. The HA binding domain in CD44 is about 160 amino acids that make up a single LINK module with extra N- and C-terminal extensions that are linked through a disulfide bond. Proteins that have these extra extensions are part of the Type B LINK module superfamily. When looking at the amino acid sequence of CD44, the amino acids that are part of the HA binding domain are found in varying locations of the amino acid sequence and this indicates that the folded protein is required for the HA binding site to come together. This differs from other HA binding proteins, such as RHAMM, which has a linear sequence for its HA binding site. It has been shown in other proteins that the LINK module superfamily interacts with HA by forming ionic bonds with carboxylic acid functional groups on HA, which is also thought to be true in CD44. There are two arginines (Arg41 and Arg 78) that are critical for ligand binding and two tyrosines (Tyr42 and Try79) that are critical for CD44 to function properly. There are a handful of other amino acids that are important for HA binding but they have been determined to be less important. CD44, found on some cell types, is non-functional and requires some form of activation for it to be able to bind to HA. The regulation of HA is very complex and could involve many different factors. The post-translational modification of glycosylation in some tissues is required for CD44 to become active as it is one of the main mechanisms that modulate CD44–HA interactions. The amount of glycosylation of the receptor differs from cell type to cell type and the activation state of CD44. When the carbohydrate at the N-terminus is removed, it causes CD44 to become active. This might indicate that certain carbohydrates block the HA binding site or that certain carbohydrates at the N-terminal lock CD44 into a non-binding conformation [129].

## 4. HABP-Based Biomaterials

Current biomaterials for HA supplementation utilize synthetic HA or HA that is derived from animal-derived resources (i.e., rooster comb, swine skin, or bovine testis). While these materials show great promise towards therapeutic and tissue regenerative uses of synthetic HA, they lack several properties of the endogenous HA, such as non-identical MWs (synthetic HA cannot reach high MWs seen in native ECM), a low degree of crosslinking or no crosslinking at all, and, finally, an exact location (i.e., pericellular or distributed homogenously through ECM) that cannot be controlled. This can lead to inflammatory responses. One method being investigated is the ability to attract more HA to the desired sight through hyaluronic acid binding peptides (HABPs). HABPs are currently being studied due to their unique ability to control HA deposition, HA localization, and HA crosslinking (Table 2). Due to their unique ability to bind to both endogenous and synthetic HA, HABPs can be used as pendants or crosslinking molecules either part of or constitutively in biomaterials. These biomaterials can be called “HA binding Biomaterials” and have significant potential, as we detail below.

### 4.1. HABP Biomaterials in Wound-Healing Research

The wound-healing process has four main phases and HA is known to help with wound closure. HA is present in the rebuilding of the ECM during wound healing. Increasing the amount of HA present in the wound-healing site is believed to aid in speeding up the healing process and decreasing scarring. The HABP is currently being researched as a peptide to help in this wound-healing process, including reducing bacterial burden at wound sites and aiding in reducing the inflammation response. Staphylococcus aureus is a systemic human pathogen that is a major contributor to surgical wound infections and treatments are difficult due to the antimicrobial resistance mechanism it has. The HABP has been shown to modulate cellular trafficking during host responses and Zaleski, K.J. et al. looked at how effective HABPs are for treating and preventing infections caused by *S. aureus*. The study looked at three different HABPs: HABP35 (LKQKIKHVVKLKVVVKLRSQLVKRKQN), HABP42 (all D-amino acids; STMMSRSHKTRSHHV), and HABP52 (GAHWQFNALTVRGGGS). HABP35 is the mouse RHAMM HA binding domain I sequence followed by the mouse RHAMM HA binding domain II sequence. The two domains are separated by a linker and contain four B-X7-B motifs. Pep-1 (HABP52) has been shown to bind to HA with high affinity and to inhibit leukocyte adhesion to HA [161].

They used a mouse model to test the antimicrobial effects of the HABP on surgical wounds. A 1 cm incision was made in the thigh muscle of the mouse and sutured back together. The *S. aureus* was then introduced into the incision and the skin closed for 3 more days before the mice were euthanized. The peptides were injected into the thigh muscle 2 cm away from the incision site (PBS was used as the control). The study found that HABPs do not exhibit antimicrobial activity for *S. aureus* and all cultures appeared to grow more of the bacteria. The effect of all three HABP treatments showed a decrease in bacterial burden at the wound site. The HABP35 and the HABP52 both showed a significant decrease. When looking at the different levels of dosing (10, 50, or 100 µ), it was found that treatment with 50 µg or 100 µg was the most effective in reducing the bacterial burden. The mice treated with HABP35 (100 µg) showed the least amount of inflammation [161].

The potential of HABPs in inflammatory leukocyte function has also been investigated. HA is expressed abundantly in the ECM as well as on cell surfaces, including binding to CD44, RHAMM, hyaluronectin, and others. CD44 binds not only to HA but also to things like collagens, fibronectins, and heparin. The HA–CD44 interaction represents one of the multiple mechanisms by which HA and CD44 may regulate cellular activities. The role of CD44 in different cellular functions has been previously discovered but not much research has been conducted regarding the role of HA. Another study by Mummert, M.E. et al. looked at developing a novel 12-mer (GAHWQFNALTVR) peptide inhibitor of HA named Pep-1 using phage display technology and showed how it had specific binding to soluble, immobilized, and cell-associated forms of HA and how it almost completely inhibited leukocyte adhesion to HA substrates. The goal of the study was to block the function of HA in the hopes of better understanding its role [162].

HABP materials can be applicable in both cosmetics and wound healing through similar methods. The goal of Yan, L. et al. was to enhance low MW HA trans member absorption in the context of cosmetics using Pep-1 and HABPs. In this study, Pep-1 is a cell-penetrating peptide that is known to permeate various substances across cellular membranes without covalent binding. Currently, microneedling is the main therapeutic for helping the HA get deeper into the skin past the stratum corneum layer [166].

Cell-penetrating peptides (CPPs) are a novel approach for the transdermal delivery of biopharmaceutical macromolecules and this study looks into its ability to increase the transmembrane and transdermal absorption of HA below the stratum corneum layer when combined with HABP.

The combination of Pep-1 and HABP could potentially promote the transmembrane efficiency of HA. The HABP was introduced to the Pep-1 to act as a linker between the Pep-1 and HA because Pep-1 cannot directly promote the absorption of HA. When tested in vitro on HaCat cells, a 1:1 ratio of HABP and Pep-1 had the best transmembrane transport ability and was further confirmed by an in vivo test on mice. The ELISA assay demonstrated that the binding ability of different concentrations of HABP and HA were linearly related and these data all together indicate that the HABP might directly bond with HA and act as a bridge between Pep-1 and HA and then confirmed that more HA was able to enter the HaCat cells with the use of both peptides. The mixture was also tested on the skin of mice for adverse reactions and none was found. Overall, the HABP and Peg-1 mixture could be a beneficial solution to allowing better transmembrane and transdermal transport of HA into the deeper levels of the dermis for cosmetology and dermatology products without the need for microneedles [166].

### 4.2. HABP Biomaterials in Musculoskeletal Research

HMWHA is a contributor to the viscoelasticity of synovial fluid. With osteoarthritis progression, HMWHA is degraded into proinflammatory LMWHA by hyaluronidases. Visco supplementation is a common treatment where HWMHA is supplemented into the synovial fluid of arthritis patients, but the longevity of the treatment is not high. Many groups are now studying implementing HABPs into different biomaterial models to aid in supporting the regeneration or repair of cartilage, preventing the progression of osteoarthritis and its ability to help with synovial fluid secretion.

One study conducted by Faust, H. et al. looked at utilizing a biomimetic system where the HABP is non-covalently bound to the cartilage surface through the heterobifunctional poly (ethylene glycol) (PEG) chain and a collagen-binding peptide (COLBP). From a mouse cartilage injury model, it was found that the HABP2-8-arm PEG-COLBP reduces cartilage degeneration without HA supplementation (Figure 5). Overall, the HABP2-8-arm PEG-COLBP increased the aggrecan expression to the healthy levels found in joints, decreased pain, and reduced cartilage deterioration, as measured by the OARSI scoring. Multiple injections were not found to be significantly more effective in treating inflammation levels, pain, or cartilage deterioration. The study was also repeated in older mice and showed that HABP2-8-arm PEG-COLBP reduces OA progression in older mice [16].

Three different HABP sequences and different PEG shapes were tested to optimize in vitro binding to HA and looked at in vivo for therapeutic effects. The HABP2-8-arm PEG-COLBP was chosen for in-depth testing of tissue localization in vivo against three different control groups: HABP2-8-arm PEG, 8-arm PEG-COLBP, and HABP2-8-arm PEG-scrambled COLBP (Figure 5) [16].

HABPs have been studied for their therapeutic effects on articular cartilage loss due to trauma and disease. In a study conducted by Unterman, S.A. et al., they looked at a poly (ethylene glycol) hydrogel with noncovalent HA binding capabilities and evaluated its ability to support cartilage formation in vitro and in an articular defect model [163]. The carboxylate group of glucuronic acid allows for relatively facile crosslinking and chemical modification of HA to form hydrogels or sponges. It is unclear how crosslinked HA scaffolds would affect cellular behavior compared to the natural presentation of HA. The goal of this study was to develop a theory to repair articular cartilage loss due to trauma or disease. To obtain a more natural presentation of HA, a HA-interacting hydrogel scaffold that noncovalently binds HA was designed. In this study, Unterman, S.A. et al. conjugated a HABP to a synthetic hydrogel scaffold based on poly(ethylene glycol) diacrylate (PEGDA). The HABP sequence used was GAHWQFNALTVR and the scrambled sequence was WRHGFALTAVNQ. The release of the HA from the PEG hydrogel depended on the initial loading dose of HA. To optimize the HA, doses loaded ranged from 0–20 mg/mL before polymerization. It was found that high concentrations of HA encapsulated into the hydrogels had minimal release of HA and it was believed that this was due to the entanglement of HA with the PEG network. When loaded with 1 and 5 mg/mL of HA, the release was much quicker [163].

Based on the load study, the PEG hydrogels were conjugated with a HABP or scrambled HABP and loaded with 5 mg/mL HA. Goat-bone-marrow-derived MSCs were selected as the cell type for in vitro testing. Cartilage formation by MSCs improved in HA-interacting scaffolds, as seen in the biochemical content, gene expression, and histological analysis. It was found that the swelling ratio did not directly depend on time or the amount of HA in the hydrogel. HA binding hydrogels increased cartilage production, as seen by higher GAG levels. It was also found that after 6 weeks, the HA binding PEG hydrogels produced significantly more Type II collagen compared to the controls [163].

HABP-based materials can also be used in conjunction with HA to facilitate and lengthen the HA in articular cartilage. It is known that lubricin, surface-active phospholipids, and HA all contribute to joint lubrication, specifically in the synovial fluid. Singh, A. et al. looked at the normal tissue interface and applied the lubrication ideas from industrial applications to create a biomimetic system for tissues and biomaterial surfaces that would work synergistically with fluid-phase biological lubricants. This was conducted by using HA binding-peptides (HABPs) covalently and non-covalently bound to surfaces through a hetero-bifunctional poly(ethylene glycol) (PEG) chain. The non-covalent chain bound HA to the modified surface, with it being endogenously available in the local fluid environment or provided exogenously. This binding of HA to the cartilage tissue mimics the role of lubricin [167].

For in vivo testing, a mixture of the HABP polymer, which was designed to target Type II collagen, and fluorescently labeled HA was injected into rat knees. This study found that combining surface treatment and the HA injection increased the longevity of linear HA in the joint. When compared to the control-bound HA, the HABP was retained for longer times in normal rat knees even though lubricin was present in the joints tested [167].

Hydrogels are a commonly used material to modulate HA towards regeneration. However, not much is known about the mechanism that bonds and retains HA in HABP hydrogels. There are many different applications for HA in the biomedical field, including HA hydrogel films being applied to full-thickness wounds and scaffolds formed from Hayff (benzyl ester derivatized HA), which have been used for skin, cartilage, nerve, and vascular tissue engineering. In their research, Elder, R. et al. look at incorporating the HABP into PEG hydrogels and studying the mechanism that binds and retains HA. HA fragment size has been shown to influence tissue synthesis and low-molecular-weight HA oligomers are known to shift HA from being noninflammatory to pro-inflammatory. Elder, R. et al. found that the higher amount of f-HA loaded into the gels containing the HA binding peptide of the scrambled HABP compared to what was loaded in PEG-only hydrogels suggests an interaction between the peptide and the HA. They also found that GAG loading was significantly reduced in the hydrogels containing the control peptide or no peptide compared to the HABP and scrambled HABP. Both simulation and experimental results showed that HABPs bind negatively charged GAGs largely through electrostatic interactions, but interactions typically occurred in a physiological environment. A simulation was run, which confirmed electrostatic interactions between the HABP and HA and provided insight into the existence of nonionic HA–peptide interactions [165]. Tissue synthesis was then looked at by measuring sGAGs through biochemical assays and it was found that the incorporation of HA improves sGSG content in the hydrogel while minimizing its loss from the hydrogel indicating and improvement in neocartilage deposition [165].

Similarly, not much is known regarding the influence of the extent and type of HA modifications on its binding to CD44 when HA is used as a chemical-modified macromer for crosslinking in hydrogels. Kwon, M.Y. et al. attempted to study this idea through novel techniques. Cells interact with HA via surface receptors, including CD44. The interaction between CD44 and HA occurs in the HA binding domain and it is understood that CD44 plays a critical role in pericellular matrix assembly, organization, and retention. HA macromers with chemical modifications that permit crosslinking are typically synthesized for fabricating hydrogels using HA. This study focused on HA hydrogels and their use in the repair of cartilage tissues; for this reason, they chose to utilize mesenchymal stromal cells (MSCs). The MSCs were also chosen for their expression of CD44 as their primary receptor for HA. In this study, they modified HA with either norbornenes (NorHA) or methacrylates (MeHA) to various extents and further characterized the hydrogels and their effects on HA downstream and PEG-diacrylate was used as the control. They were able to develop many novel methods to help quantify the interactions of CD44 with modified HA, such as the use of atomic force microscopy (AFM) [168]. The low- and medium-modified groups contained comparable concentrations of glycosaminoglycans compared to reported values of healthy human articular cartilage [168].

Previous studies utilized the HABP GAH conjugated to polymers to inhibit the progression of OA but Deloney, M. et al. modified an anionic, polymeric, hollow nanoparticle (hNP) composed of N-isopropyl acrylamide (NIPAm), 2-acrylamido-2-methyl-1-propansulfonic acid (AMPS), N,N’-Bis(acryloyl) cystamine (BAC), and acrylic acid (AAc) with the HABP GAH to mimic aggrecan function and generate a HA binding nanoparticle (GAH-hNP). This study supports the idea that anionic hNP conjugated with GAH will restore the compressive stiffness of aggrecan-depleted cartilage and inhibit further degradation of ECM. It was found that by increasing the ratio of GAH to AAc, the number of GAH peptides per nanoparticle increased significantly. When looking at HA binding and diffusion into cartilage explants, all concentrations of 19 GAH-hNP were significantly bound to HA, as determined by dynamic velocity (DV) measurements. Treating cartilage explants with trypsin has been shown to strip aggrecan from the explant without damaging the chondrocytes, HA, or collagen, making it a good ex vivo model for OA. The stiffness of bovine explant models was measured and compared to healthy cartilage stiffnesses. Overall, there was a significant improvement in compressive stiffness with the GAH-hNP, which indicates the importance of HA binding similar to that seen with aggrecan. A decrease in compressive stiffness between Days 6 and 12 in the GAH-hNP treated cartilage suggested that some of the particles are lost from the tissue either via diffusion or by degradation of the particles. Fragmented collagen and HA are known catabolic stimulants so dampening them may help slow OA progression. It is also believed that these GAH-hNP may help in keeping the chondrocyte-to-chondrocyte communication that can be lost in OA progression due to the breakdown of the ECM [169].

### 4.3. HABP Biomaterials as Biological Lubrication Agents

Singh et al. also looked at the frictional components since the physical lubrication properties of HA are therapeutically relevant to the joint’s function. The static and kinetic total-friction values for the cartilage tissue decreased significantly when tested in a HA bath. The HABP was able to consistently reduce the total friction values (Figure 6). The study went on to test the HABP effects on osteoarthritic cartilage samples and found that when treated with the HABP coating that produced surface-bound HA, the static and friction values nearly equaled those samples that were placed in the HA bath. Singh, A. et al. also discussed the possible ocular application for the HABP and how preliminary testing found that, when utilizing the eye-drop method, the HABP functionalized to the eye surface without damaging the epithelial layer and could recruit and retain HA [167].

### 4.4. HABP Biomaterials in Cancer Research

Many of the cellular functions associated with HABPs, like migration, adhesion, growth, differentiation, and apoptosis, are also important in the growth of tumors. It has been found in previous studies that proteins that bind with HA can also inhibit the growth of tumor cells. One of these proteins is called metastain, which consists of a fragment of aggrecan and the LINK protein that binds to HA with high affinity and has been shown to inhibit the growth of tumor cells. Similarly, soluble CD44, RHAMM protein and factor TSG-6 are known to bind with HA and have been shown to inhibit tumor growth.

Lui, N. et al. identified a HABP they named P4 that inhibited tumor growth cells in tissue cultures as well as on chorioallantoic membranes of chicken embryos. Through various tests, it was found that the P4 peptide had a strong ability to bind to HA. When tested in nude mice, it was found that the tumor cells transfected with P4 grew slower. It was shown in further studies that apoptosis was induced by P4 peptides rapidly entering the cells where they interact with the Bcl-2 family of proteins. Lui, N. et al. concluded that their results suggest that proteins and peptides that can bind to HA may be a new compound category to turn to for antitumor activity [170].

Xu, X. et al. similarly studied the biological activity of a 42-amino-acid peptide (HABP) that contained three HA binding motifs (BX_7_B) from the human brain HA binding protein. In their initial studies, it was found that the synthetic HABP inhibited tumor cell proliferation and colony formation. The growth of tumors was also blocked on the chorioallantoic membranes of 10-day-old chicken embryos. It was noted that the embryo development did not appear to be negatively affected, which indicated that the growth of the other cell types was not affected. The group then transfected mda-435 melanoma cells with an expression vector for the HABP and it was found that these cells grew slower in nude mice than in non-transfected cells. The study indicated that the apoptosis of tumor cells was triggered by the HABP activating caspase-8, caspase-3, and poly(ADP-ribose) polymerase [171].

### 4.5. HABP Biomaterials in Tissue Engineering

The ECM modulates many cell functions and this regulation is provided by key ECM components forming a complex network. Engineered surfaces have also been applied to control cell phenotypes, like protein nanosheets assembled at the oil-water interface that were used as a culture system. Differential integrin expression was shown to modulate cell adhesion and spreading to patterned nanofibers with controlled diameters. HA significantly contributes to the development and progression of many pathological conditions, such as osteoarthritis or dry eye. HMW HA is known to inhibit the proliferation and migration of vascular endothelial cells and is also known to be immunosuppressive and anti-inflammatory. Surfaces enabling the immobilization of HA without chemical modification would provide a more biomimetic presentation of HA to elucidate its interactions with cells.

Pang, X. et al. had previously developed a 2D model surface consisting of HA immobilization on self-assembled monolayers (SAMs) of a HA binding peptide (Pep-1). The study’s aim was to investigate the influence of HA size on endothelial cell motility and analyze single-cell adhesion and migration in the presence of HA immobilized or in the soluble form. Thiolated Pep-1 included the HABP sequence GAHWQFNALTVR. The first step in the study was to characterize the mixed and Pep-1 SAMs. SAMs with low (1%) and high (100%) densities of Pep-1 on Au-coated substrates were created. There were differences in thickness that were found between dry and wet conditions on both mixed and Pep-1 SAMs with or without immobilized HA. There were great thickness increases in the wet conditions. The contact angle was used to characterize the hydrophobicity of mixed and Pep-1 SAMs with or without HA. Mixed SAMs were noticeably more hydrophilic and the deposition of HA on Pep-1 SAMs also resulted in similar hydrophilic surfaces [164].

Low-molecular-weight (LMW) HA immobilized via Pep-1 SAMs facilitating cell spreading and single-cell migration. Human umbilical vein endothelial cells (HUVECs) were seeded on Pep-1 SAMs with or without HA and the viability was assessed on Days 2, 5, and 7. Viabilities were high on all surfaces and time points. HMW HA slightly inhibited cell spreading. The LMW HA immobilized via Pep-1 SAMs accelerated endothelial cell alignment and elongation under laminar flow conditions. Directional migration was regulated by LMW HA immobilized via Pep-1 SAMs. There was a speeding-up of directional cell migration suggesting that endothelial cells were moving towards LMW HA. CD44 expression was measured through fluorescent intensity and there was no significant difference between the groups when cells were seeded on immobilized HA. The LMW HA immobilized via Pep-1 SAMs enhanced the assembly of focal adhesions. The HUVECs seeded on 5 kDa HA-immobilized surfaces displayed higher levels of F-actin and vinculin than the 700 kDa HA-coated surfaces. They propose that the increase in cell motility from LMW HA is a result of the differential recruitment of vinculin at FAs. They finished by discussing the hope of translating these results into therapeutic applications. They mentioned using Pep-1 SAMs for the simple and effective immobilization of LMW HA on implantable endovascular devices, such as artificial vascular grafts and stents, to promote rapid endothelialization and improve their performance [164].

## 5. Future Potential of HA Binding Biomaterials

In this paper, we first discussed the role of HA in the development of various tissues are their function within the tissue. Then we went on to talk about the different therapeutic effects of HA in these tissues. Next, the proteins and receptors were broken down and the effects of these bindings were discussed. Finally, we looked into HABPs and how they can be used in biomaterials for healing and lubrication in the body. To date, modulating the crosslinking state, precise location, and physiologically relevant MW of HA is a significant challenge in HA-based biomaterials applications. We believe that HA binding biomaterials can not only mitigate the challenges above but also provide a new class of biomaterials where the crosslinking of HA, proteoglycans (i.e., aggrecan, versican), and HA receptors (i.e., CD44, RHAMM, or LYVE-1) can be engaged towards the desired biological outcome. Due to their short sequences and modality, HABPs can be combined with other peptide sequences, such as collagen mimetic peptides, to obtain dual functions. Using surface conjugation techniques, many synthetic materials could become Hyaladherin by functionalizing their surfaces. In Figure 7 we depict a possible vision where HA-binding biomaterials can be formed (i.e., nanofibers, particles, hydrogels, and hybrid systems) for leveraging the regenerative properties of HA. One important application of HA binding biomaterials is preventing fibrotic capsule (FC) formation around implanted biomaterials and devices. Since high levels of HA deposition in wound beds correlate with scarless and antifibrotic healing, we anticipate that HABP interfaces around implanted devices can help overcome the FC formation. Secondly, HA binding particle systems can be used to target tumor sites (as it is known that LMWHA is rich in the tumor stroma) or lymphatic systems (via engaging with LYVE-1 receptors). Owing to their high surface-to-volume ratio, HA binding biomaterials can also be used to treat osteoarthritis (OA). HA binding hydrogels can be attractive tools for soft tissue repair, such as skin wound healing, neural regeneration, corneal repair, and reproductive-system-related conditions. Finally, hybrid systems combining hydrogels and particles could be developed towards multifunctional and hierarchically assembled HA binding biomaterials for cardiovascular applications (eliciting angiogenesis for tissue engineering) and corneal repair via drug-releasing particle-embedded hydrogels. We believe that this new class of biomaterials will give the researcher the advantage of utilizing endogenous, physiologically relevant, and crosslinkable HA, which could elicit biologically relevant experimental outcomes.

## Figures and Tables

**Figure 1 biomimetics-09-00499-f001:**
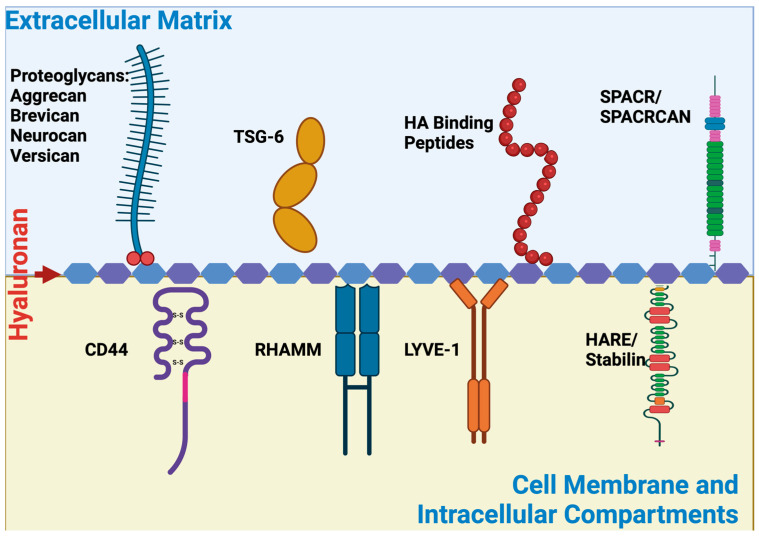
**Diverse HA interactions in the cellular microenvironment.** Diagram showing HA interactions with common receptors, such as CD44, RHAMM, LYVE-1, and HARE/Stabilin as transmembrane receptors. Proteoglycans, such as aggrecan, brevican, neurocan, and versican, bind with HA through LINK domains at the ECM. The other HABPs, such as TSG-6, HA binding peptides, and SPACR/SPACRCAN, bind to HA through other protein-protein interactions.

**Figure 2 biomimetics-09-00499-f002:**
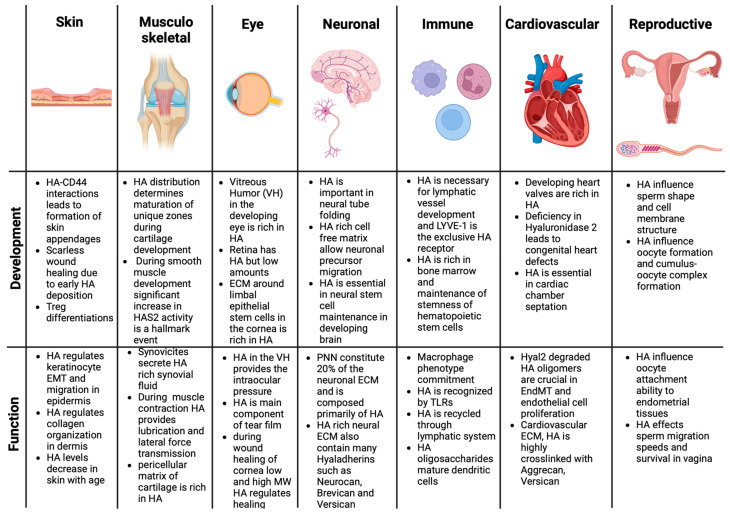
**Involvement of HA in the development and function of HA-rich tissues.** Skin, musculoskeletal, eye, neuronal, immune, cardiovascular, and reproductive tissues are some of the main tissues whose development and function are tightly regulated by HA.

**Figure 3 biomimetics-09-00499-f003:**
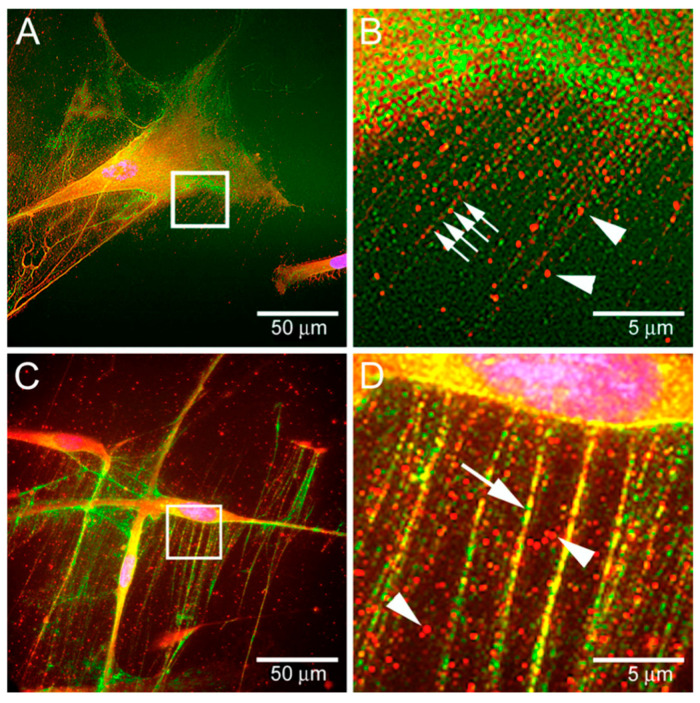
Localization of HA (red) and versican (green) at the pericellular space. (**A**,**B**) are the control cells and you can see the HA binding along single versican strands. The arrows point to the HA alignments and the boxes are higher magnification of that area. (**C**,**D**) are the treated surfaces [101].

**Figure 4 biomimetics-09-00499-f004:**
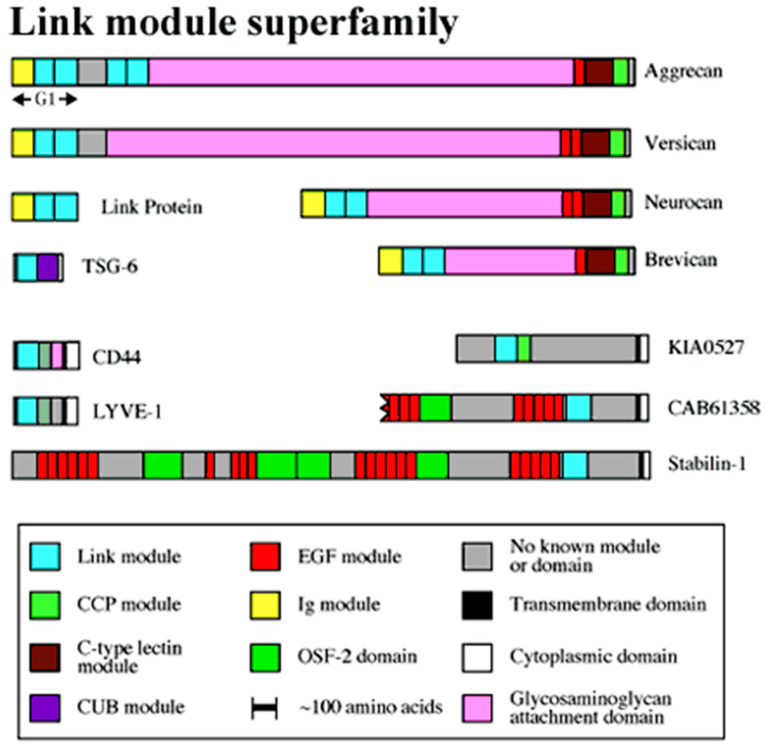
LINK modules are present in a variety of HA binding proteins and their location determines the extent of the HA interactions they will undergo [3].

**Figure 5 biomimetics-09-00499-f005:**
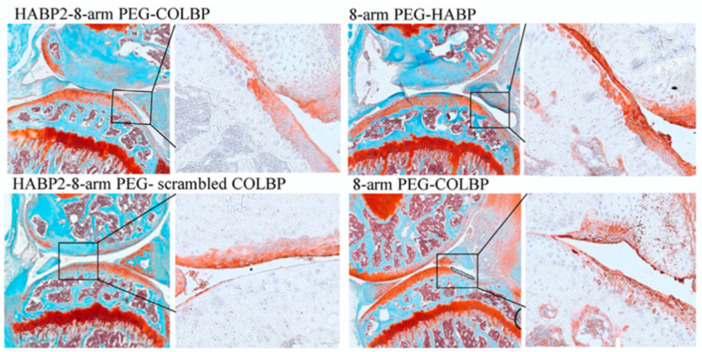
**A total of** 24 h after the injection, the cartilage was imaged. The full construct retained peptides at the cartilage and synovium. This further highlighted that the HABP2-8-arm Peg-COLBP was the best choice for retaining HA in the cartilage and synovium [16].

**Figure 6 biomimetics-09-00499-f006:**
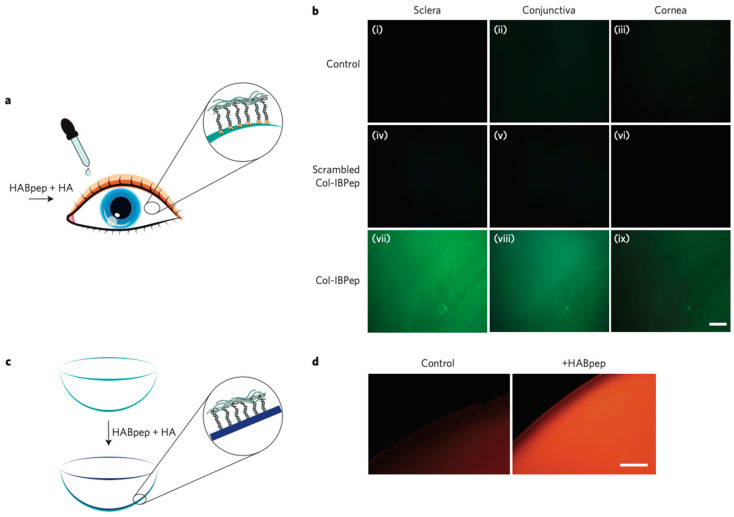
Ocular surface application of the HABP polymer system. In these images, (**a**) demonstrates the application of the HABP to the eye as an eye-drop solution with the hope of retaining HA on the eye surface; (**b**) fluorescence images of HA demonstrate the higher levels of HA present when the Col-1B peptide (i–iii) was used compared to the scrambled peptide (iv–vii) and the control (vii–ix); (**c**) is the idea that the HABP eye-drops could be modified into a contact lens by covalently reacting the peptides; and (**d**) this image shows the increased HA-rhodamine retention on the surface when the HABP is used compared to the control [167].

**Figure 7 biomimetics-09-00499-f007:**
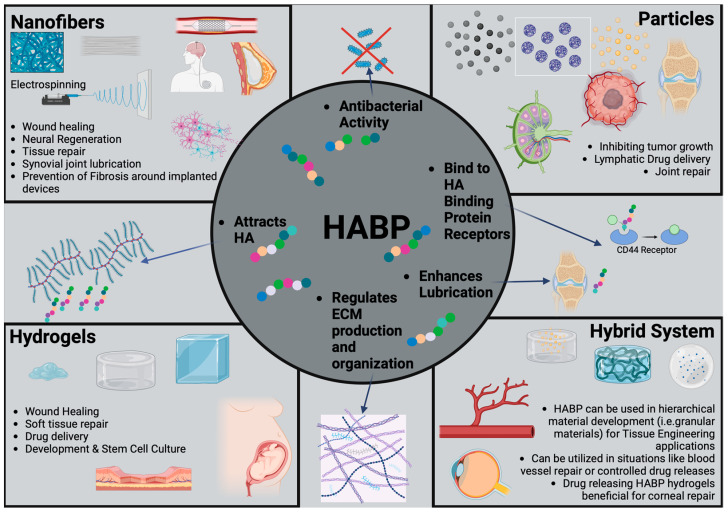
HABPs can be integrated into multiple biomaterials, such as particles, nanofibers, hydrogels, and hybrid systems. These HABPs have been shown to attract HA, bind HA to binding protein receptors, enhance lubrication, regulate ECM production and organization, and have antibacterial activity. Each biomaterial also has been shown to be helpful in specific types of treatments.

**Table 1 biomimetics-09-00499-t001:** Functions of HABPs. Information is provided based on the PUBMED NCBI GeneBank ID Database with staying respectful to the original information. ** is based on RNA sequencing data and normalizes read count based on gene length and the total number of mapped reads*.

Protein	NCBI Gene ID	Function	Expression in Normal Tissues (Reads per Kilobase of Transcript per Million Mapped Reads) *
Aggrecan	176	An integral part of the ECM in cartilaginous tissue and it withstands compression in cartilage Enables carbohydrate binding Enables ECM structural constituent conferring compression resistance [101,102] Enables HA binding Enables metal ion binding Enables protein binding [103]	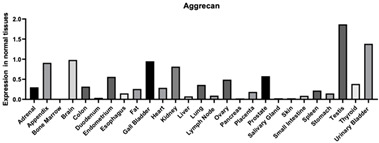
Neurocan	1463	Chondroitin sulfate proteoglycan that is involved in the regulation of cell adhesion and migration Enables calcium ion binding Enables carbohydrate binding Enables HA binding [104]	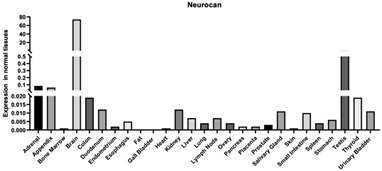
Versican	1462	The protein encoded is a large chondroitin sulfate proteoglycan and is a major component of the extracellular matrix. This protein is involved in cell adhesion, proliferation, migration, and angiogenesis and plays a central role in tissue morphogenesis and maintenance. Enables calcium ion binding Enables carbohydrate binding Enables ECM structural constituent conferring compression resistance [102,105,106,107] Enables glycosaminoglycan binding [108] Enables HA binding [108] Enables protein binding [109]	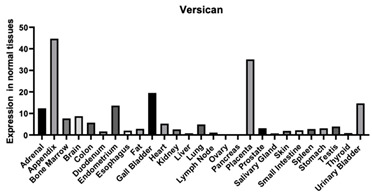
Brevican	63827	This gene encodes a member of the lectican family of chondroitin sulfate proteoglycans that is specifically expressed in the central nervous system. May function in the formation of the brain’s extracellular matrix Enables carbohydrate binding Enables HA binding Enables protein binding	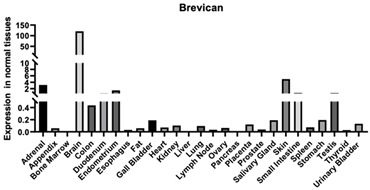
TSG-6	7130	A secretory protein that contains a hyaluronan binding domain. This protein has been shown to form a stable complex with inter-alpha-inhibitor (I alpha I), and thus enhance the serine protease inhibitory activity of I alpha I, which is important in the protease network associated with inflammation. Enables calcium ion binding [110] Enables carboxylesterase activity Enables fibronectin binding [110] Enables HA binding [111] Enables protein binding [111,112,113,114,115]	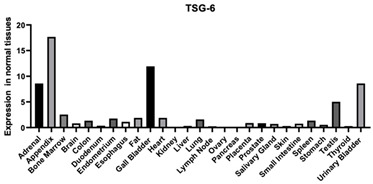
CD44	960	CD44 is found in a variety of tissues with the highest expression found in the appendix and the skin A cell-surface glycoprotein is involved in cell-cell interactions, cell adhesion, and migration. It is a receptor for HA and can also interact with other ligands, such as osteopontin, collagens, and matrix metalloproteinases (MMPs). This protein participates in a wide variety of cellular functions including lymphocyte activation, recirculation and homing, hematopoiesis, and tumor metastasis. Enables collagen binding [116] Contributes to cytokine receptor activity Enables HA binding [117,118,119] Enables protein binding [120] Enables transmembrane signaling receptor activity	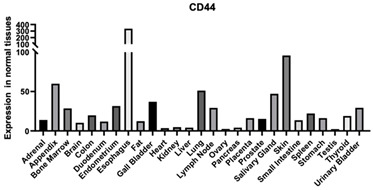
LYVE	10894	A type I integral membrane glycoprotein. The encoded protein acts as a receptor and binds to both soluble and immobilized hyaluronan. This protein may function in lymphatic hyaluronan transport and have a role in tumor metastasis. Enables cargo receptor activity Enables HA binding [121] Enables protein binding Enables signaling receptor activity Enables transmembrane signaling receptor activity	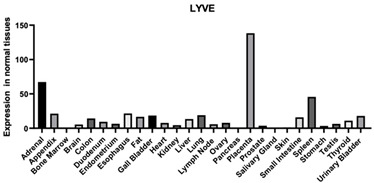
RHAMM	3161	Involved in cell motility. It is expressed in breast tissue and together with other proteins, it forms a complex with BRCA1 and BRCA2, thus is potentially associated with a higher risk of breast cancer. Cargo receptor activity—binding to a specific substance to deliver it to a transport vesicle HA binding [122] Protein binding [63,64]	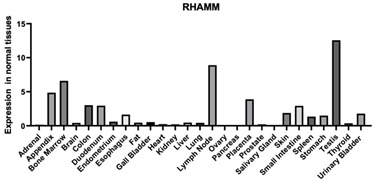
Stabilin	55576	This protein is a large, transmembrane receptor protein that may function in angiogenesis, lymphocyte homing, cell adhesion, or receptor scavenging. The protein contains 7 fasciclin, 15 epidermal growth factor (EGF)-like, and 2 laminin-type EGF-like domains as well as a C-type lectin-like hyaluronan binding LINK module. The receptor has been shown to bind and endocytose ligands such as hyaluronan, low-density lipoprotein, Gram-positive and Gram-negative bacteria, and advanced glycosylation end products. Enables calcium ion binding Enables HA binding Enables low-density lipoprotein particle binding [123] Enables protein binding [124] Enables protein-disulfide reductase activity [123] Enables scavenger receptor activity [118]	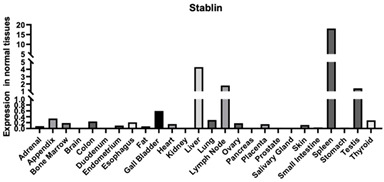
SPACR	3617	A protein that is a major component of the retinal interphotoreceptor matrix. The encoded protein is a proteoglycan that is thought to play a role in maintaining the viability of photoreceptor cells and in the adhesion of the neural retina to the retinal pigment epithelium. Enables chondroitin sulfate binding Enables extracellular matrix structural constituent [125] Enables heparin binding Enables HA binding	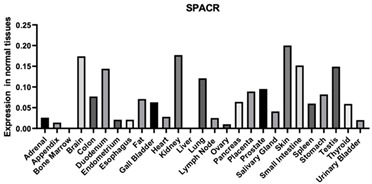
SPACRCAN	50939	This protein binds chondroitin sulfate and hyaluronan and is a proteoglycan. The encoded protein plays a role in the organization of the interphotoreceptor matrix and may promote the growth and maintenance of the light-sensitive photoreceptor outer segment. Enables ECM structural component [126] Enables heparin binding Enables HA binding	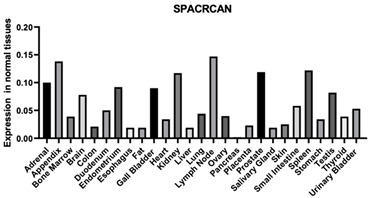
PHBP HA binding peptide 2	3026	A member of the peptidase S1 family of serine proteases. The encoded preproprotein is secreted by hepatocytes and proteolytically processed to generate heavy and light chains that form the mature heterodimer. Further autoproteolysis leads to smaller, inactive peptides. This extracellular protease binds hyaluronic acid and may play a role in the coagulation and fibrinolysis systems Enables calcium ion binding Enables GAG binding Enables peptidase activity [127] Enables serine-type endopeptidase activity	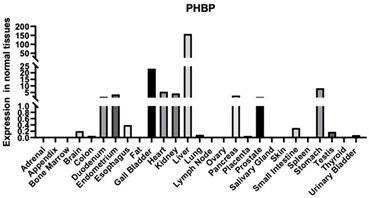

**Table 2 biomimetics-09-00499-t002:** HABP Sequences from currently published research and the outcome of each version..

HABP Sequence	Results
HABP35 LKQKIKHVVKLKVVVKLRSQLVKRKQN	Decease bacterial burden at the wound site [161]
HABP42 (all D-amino acids; STMMSRSHKTRSHHV)	Decease bacterial burden at the wound site [161]
GAHWQFNALTVRGGGS (HABP52)	Decease bacterial burden at the wound site [161] Binds to HA with high affinity and inhibits leukocyte adhesion to HA
GAHWQFNALTVR	High HA binding, inhibited the adhesion of leukocytes and inhibited hypersensitivity responses [162] HA interactive PEG hydrogels increased the cartilage tissue production in the defects (reduced cartilage degradation) [163] More hydrophilic and attracted more HA than other SAMs [164]
TLRAIWPMWMSS (“Pep-4”)	Not successful in HA binding [162]
IPLTANYQGDFT (“Pep-5”)	Not successful in HA binding [162]
(TSYGRPALLPAA “Pep-2”)	Not successful in HA binding [162]
(MDHLAPTFRPAI “Pep-3”)	Not successful in HA binding [162]
RYPISRPRKRC	measuring sGAGs through biochemical assays and it was found that the incorporation of HA improves sGSG content in the hydrogel while minimizing its loss from the hydrogel indicating an improvement in neocartilage deposition [165]
GYPISGPGGGC (charge control peptide)	HABP bind negatively charged GAGs largely through electrostatic interactions, but interactions typically occur in a physiological environment [165]
WRHGFALTAVNQ (scrambled)	HABP bind negatively charged GAGs largely through electrostatic interactions, but interactions typically occurred in a physiological environment [120]
CNGRCGGKQKIKHVVKLKVVVKLKSQLVKRKVVVRRRKKIQGRSKR	the tumor cells transfected with P4 grew slower [127]

## Data Availability

Not applicable.

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
