# Peer review of "Biomimetic Hyaluronan Binding Biomaterials to Capture the Complex Regulation of Hyaluronan in Tissue Development and Function"

_biomimetics, 2024, doi:10.3390/biomimetics9080499_

Round 1

Reviewer 1 Report

Comments and Suggestions for Authors

The authors did a commendable job of covering the literature comprehensively within different therapeutic areas highlighting the role of HA in Tissue development and function.

I have minor comments for the authors to address:
1) Section 2 in which authors comprehensively discuss the role of HA in tissue development and function; the authors are recommended to shorten the section, highlight key works in each sub-section, and refer to readers different review papers in the particular field. 

2) Similarly for section 3 in which authors discuss HA binding protein and the discovery of HA binding peptides, authors are again recommended to summarize the section in a couple of pages in addition to the table and refer to the review papers for the particular HA binding proteins in case readers are more interested in learning about it.

3) For section 4, development of HABP biomaterials, authors are recommended to add their recommendations on what are the future areas of interest for designing the HABP biomaterials. While reviews do a great job of summarizing the fields, reviews also guide researchers toward new areas/ideas. The authors are in a great position to make those recommendations given the comprehensive literature authors have covered.

4) Authors should further expand on the conclusions and future directions section. The current text just summarizes what each section is about. Authors should discuss the highlights from each section in the conclusion section and add more details on the future directions. Currently, no statements are provided regarding future directions.

Author Response

Reviewer 1 Report:

Comments to authors

Overview:

The authors did a commendable job of covering the literature comprehensively within different therapeutic areas highlighting the role of HA in Tissue development and function.

General Comments:
- Section 2 in which authors comprehensively discuss the role of HA in tissue development and function; the authors are recommended to shorten the section, highlight key works in each sub-section, and refer to readers different review papers in the particular field.

Response:
The highlighting key works is beyond the scope of the work. We added referrals to readers for different review papers.

-Similarly for section 3 in which authors discuss HA binding protein and the discovery of HA binding peptides, authors are again recommended to summarize the section in a couple of pages in addition to the table and refer to the review papers for the particular HA binding proteins in case readers are more interested in learning about it.
Response:

Added in referrals to review papers.

-For section 4, development of HABP biomaterials, authors are recommended to add their recommendations on what are the future areas of interest for designing the HABP biomaterials. While reviews do a great job of summarizing the fields, reviews also guide researchers toward new areas/ideas. The authors are in a great position to make those recommendations given the comprehensive literature authors have covered.
Response:

-Authors should further expand on the conclusions and future directions section. The current text just summarizes what each section is about. Authors should discuss the highlights from each section in the conclusion section and add more details on the future directions. Currently, no statements are provided regarding future directions.

Response:

A new paragraph added at the end of the manuscript based on the suggestions. We appreciate your constructive critique.

Reviewer 2 Report

Comments and Suggestions for Authors

1- Hyaluronan plays an effective role in the development and functioning of the reproductive system in women. Because it has been proven that the binding of hyaluronan and CD44 during cumulus expansion is essential for maturation-promoting factor (MPF) activity and oocyte meiosis. Also, Hyaluronan regulates sperm-induced inflammation. Hence, adding a section entitled "Role of Hyaluronan in reproductive system development and function" seems necessary. Moreover, several studies have reported that HA-rich medium improved embryo growth and implantation in reproductive-assisted technology. Using the following resources can greatly help in discussing the function of hyaluronan in the reproductive system, so please cite the following and discuss them.

https://onlinelibrary.wiley.com/doi/full/10.1002/adbi.202300621

https://link.springer.com/article/10.1007/s43032-021-00520-7

2- Because many findings have evidenced the antioxidant properties of hyaluronan. Also, its role in regulating the immune system. Please discuss the effects of hyaluronan in treating inflammatory diseases of the genital tract, such as varicocele.

3- Because there is emerging data that hyaluronan and its degradation products have an important role in the pathobiology of the respiratory tract. Therefore, it seems necessary to add a section entitled"Role of Hyaluronan in Respiratory Tissue Development and Function "to complete the article.

Author Response

Reviewer 2 report:

Comments to authors:

Hyaluronan plays an effective role in the development and functioning of the reproductive system in women. Because it has been proven that the binding of hyaluronan and CD44 during cumulus expansion is essential for maturation-promoting factor (MPF) activity and oocyte meiosis. Also, Hyaluronan regulates sperm-induced inflammation. Hence, adding a section entitled "Role of Hyaluronan in reproductive system development and function" seems necessary. Moreover, several studies have reported that HA-rich medium improved embryo growth and implantation in reproductive-assisted technology. Using the following resources can greatly help in discussing the function of hyaluronan in the reproductive system, so please cite the following and discuss them.

https://onlinelibrary.wiley.com/doi/full/10.1002/adbi.202300621

https://link.springer.com/article/10.1007/s43032-021-00520-7

Response:

We have added a section “Role of HA in Reproductive System Development and Function”. We included both of the articles provided.

2- Because many findings have evidenced the antioxidant properties of hyaluronan. Also, its role in regulating the immune system. Please discuss the effects of hyaluronan in treating inflammatory diseases of the genital tract, such as varicocele.
Response:

This was addressed in the added section of Role of HA in Reproductive system development and function.

3- Because there is emerging data that hyaluronan and its degradation products have an important role in the pathobiology of the respiratory tract. Therefore, it seems necessary to add a section entitled "Role of Hyaluronan in Respiratory Tissue Development and Function "to complete the article.

Response:

We have added a section “Role of HA in Respiratory Tissue Development and Function”.

Round 2

Reviewer 2 Report

Comments and Suggestions for Authors

the manuscript has been revised well and can be considered for publishing.